# CXCL13 as a Biomarker: Background and Utility in Multiple Sclerosis

**DOI:** 10.3390/biom14121541

**Published:** 2024-11-30

**Authors:** Andrew R. Pachner, Steven Pike, Andrew D. Smith, Francesca Gilli

**Affiliations:** Geisel School of Medicine at Dartmouth, Dartmouth-Hitchcock Medical Center, Lebanon, NH 03756, USAandrew.d.smith.iii@hitchcock.org (A.D.S.); francesca.gilli@dartmouth.edu (F.G.)

**Keywords:** chemokine, chemoattractant, CXCL13, multiple sclerosis, Lyme neuroborreliosis, ectopic lymphoid follicle, point-of-care testing, ELISA, Luminex, Olink, matrix effects

## Abstract

CXCL13 is a chemokine which is upregulated within the CNS in multiple sclerosis, Lyme neuroborreliosis, and other inflammatory diseases and is increasingly clinically useful as a biomarker. This review provides background for understanding its function in the immune system and its relationship to ectopic lymphoid follicles. Also reviewed are its utility in multiple sclerosis and Lyme neuroborreliosis and potential problems in its measurement. CXCL13 has the potential to be an exceptionally useful biomarker in a range of inflammatory diseases.

## 1. Background and Clinical Significance of Intrathecally Produced CXCL13

### 1.1. The Role of CXCL13 in Immunology

CXCL13, initially termed B cell-attracting chemokine-1 (BCA-1) or B lymphocyte chemoattractant (BLC), was first identified in 1998 [1,2] as a ligand for the receptor BLR1, later termed CXCR5. BLR1 is a protein on the surface of Burkitt’s lymphoma and B cells, and it had previously been cloned. Knockouts of BLR1 had yielded a dramatic phenotype: markedly abnormal lymph nodes with impaired migration of lymphocytes, altered lymphoid follicles, and no functional germinal centers [3]. Subsequent work demonstrated that B cells go through developmental stages within lymph nodes that involve, among other changes, decreased responsiveness to CXCL13 as they differentiate into plasma cells [4].

### 1.2. CXCL13 Production

CXCL13 can be produced by a variety of cell types, including stromal cells such as follicular dendritic cells and marginal reticular cells, as well as cells of the macrophage/monocyte lineage including monocytes, macrophages, and microglia. In Lyme neuroborreliosis (LNB), where CXCL13 production is robust, CXCL13 was shown to be produced by myeloid and plasmacytoid dendritic cells [5]. In EAE, CXCL13 has been demonstrated to be produced in the CNS, i.e., intrathecal production of CXCL13 (IP_CXCL13_) by follicular dendritic cells in lymphoid follicle-like structures [6]. In secondary progressive MS, CXCL13 was produced in the CNS by stromal cells and FDC (follicular dendritic cells) [7]. Krumbholz et al. [8] showed that in vitro, CXCL13 was produced in humans by monocytes and at much higher levels by macrophages. Although most of the literature identifies stromal and macrophage/monocyte lineage cells as the producers of CXCL13, follicular helper T cells (Tfh) also produce CXCL13 [9,10]. This important T-cell subset, reviewed below, also responds to CXCL13 through CXCR5 on its surface.

### 1.3. CXCL13 Chemoattraction in the CNS

Kowarik et al. concluded that CXCL13 is the most important molecule for recruitment of B cells to the CSF [11], based on correlation of CSF CXCL13 concentrations to CSF B cells, plasmablasts, and intrathecal Ig synthesis. Krumbholz et al. [8] demonstrated that CXCL13 was produced in actively demyelinating multiple sclerosis lesions, but not chronic inactive lesions, and was localized in perivascular infiltrates and scattered infiltrating cells in lesion parenchyma. They also showed that about 20% of CSF CD4+ cells and almost all B cells expressed the CXCL13 receptor CXCR5.

### 1.4. Follicular Helper T Cells

Tfh is one of the populations of cells expressing CXCR5, and they are key cells in the germinal centers in the secondary lymphoid organs (SLO), such as the lymph nodes, spleen, tonsils, and gut-associated lymphoid tissue (GALT). They comprise a subset of CD4+ T cells critical for T cell-dependent B-cell activation, i.e., affinity maturation through somatic hypermutation and class-switch recombination (CSR), which ultimately leads to the formation of long-lived plasma cells and memory B cells. Like germinal center B cells, Tfh cells express CXCR5, the primary receptor for CXCL13, which helps CXCL13-directed positioning from T-cell zones into B-cell follicles in germinal centers. As lymphocyte trafficking and ectopic lymphoid follicles in MS (see below) have become intensively studied, there has been a plethora of studies on Tfh in MS [12,13,14,15,16]. Tfh both produce and respond to CXCL13.

### 1.5. CXCL13 and the Concept of Ectopic Lymphoid Follicles (ELFs)

#### 1.5.1. ELFs in Experimental Models of MS

It is likely that IP_CXCL13_ is intimately associated with the development within the CNS of organized structures of lymphoid cells called ectopic lymphoid follicles (ELFs). ELFs have been defined by Aloisi et al. [17] and Zhan et al. [18] as: “large B-cell aggregates localized primarily in the subarachnoid space, mainly inside the cerebral sulci, that display several germinal centre-like features (i.e., presence of stromal/follicular dendritic cells expressing CXCL13, B-cell proliferation, expression of activation-induced cytidine deaminase and plasma cell differentiation) but lack the typical structure of lymphoid follicles with a germinal centre and a mantle zone and contain mainly memory B cells”. While lymphoid follicles in lymph nodes and other secondary lymphoid organs have been extensively studied, ELFs, also called tertiary lymphoid structures (TLS), tertiary lymphoid tissue (TLT), or tertiary lymphoid organs (TLOs), have elicited less study, although interest has been increasing recently, and reviews of their role in MS [19,20,21] and other chronic inflammatory processes [22] have recently been published. CNS ELFs provide an environment for cellular and humoral immune responses outside of lymphoid tissues.

Experimental models in which ELFs have formed have provided data helpful in understanding their development, organization, and function. ELFs, also called meningeal ectopic lymphoid tissue (mELT), are a consistent finding in the pathology of a spontaneous chronic experimental autoimmune encephalomyelitis (EAE) model of mice with mutant T-cell and B-cell receptors specific for myelin oligodendrocyte glycoprotein (MOG) [23,24,25], and single-cell profiling studies have shown that the cell makeup of mELTs is similar to that of lymph node lymphoid follicles. ELFs have also been found in other models of EAE [6,25,26]. Columba-Cabezas et al. were able to suppress ELFs and CNS CXCL13 production in EAE using a lymphotoxin beta receptor-Ig fusion protein, and they proposed that this approach might be able to be used for the treatment of MS [27].

ELF-like aggregates have also been a feature of Theiler’s murine encephalomyelitis virus (TMEV)-induced demyelinating disease (TMEV-IDD), a virally-mediated model of MS. This model mimics progressive forms of MS in showing gradual disability accrual over time, spastic paralysis, and characteristic pathology including atrophy, neuronal, primarily axonal, injury, demyelination, remyelination, and extensive CNS inflammation [28,29,30]. Intrathecal antibody synthesis is a consistent feature of TMEV-IDD [31,32], as are inflammatory aggregates, a term that was used instead of ELFs [33]. That alternate term was used because the aggregates in TMEV-IDD lacked many of the characteristics of follicles found in SLOs, especially spatial organization into distinct T- and B-cell zones. Most of the B cells in the CNS were memory B cells and antibody-secreting cells, and B cells of various developmental stages were present in the CNS with mature and isotype-switched B cells mostly localized to the meninges and perivascular space. IgG isotype-switched B cells could be found in the parenchyma. FACS analysis of the CNS B cells showed increased expression of activation markers such as GL7, an antigen present on the surface of germinal center B cells in SLO. Importantly, CXCL13 was highly expressed, as were the costimulatory molecule CD80 and B cell-related chemokines CXCL9, 10, and 19, as well as BAFF and APRIL. Thus, it is clear that, at least in the mouse during the chronic inflammation of TMEV-IDD, CXCL13 expression can occur independently of highly organized ELFs resembling SLOs. TMEV-IDD is thus a good model of CNS compartmentalized cellular and humoral immune responses with CXCL13 expression in the CNS.

Another viral infection of mice that mimics MS in which CXCL13 is upregulated is infection with mouse hepatitis virus (MHV), as shown in a study of wild-type and CXCL13-KO mice [34]. This study demonstrated that CXCL13-KO mice could mount effective peripheral humoral responses, but they had deficient CNS accumulation of differentiated B cells. Another chemokine important in the model is CXCL10, which can be involved in the recruiting of antibody-secreting cells to the CNS; the main production of CXCL10 in this model interestingly was by astrocytes [35].

#### 1.5.2. ELFs in MS

Features of ELFs in specimens of MS brain have been described as long ago as 1979 by John Prineas [36]. More recently, investigators in Italy have been leaders in studying ELFs and CXCL13 in human MS. After initial work in EAE [6], Magliozzi and her collaborators in Rome in work published in 2004 detected ELFs and CXCL13 expression in the meninges of two out of three patients with SPMS; the ELFs contained CD138+ plasma cells and proliferating B cells [37]. In an extension of this work to a larger number of postmortem brains from SPMS patients in 2007 [7], they found that B-cell follicles were detected in the meninges in 41.4% of the SPMS cases but not in PPMS cases. They also determined that meningeal B-cell follicles were found next to subpial cortical lesions, raising the possibility that soluble factors from follicles have a pathogenic role. They also felt that the finding of CXCL13 expression was critical for the characterization of the ELFs [38]. The high levels of CXCL13 in the CSF or ELFS of MS patients, along with other proteins associated with lymphoid follicles such as activation-induced cytidine deaminase (AICDA) [39] and proliferating B cells, such as Ki67+ centroblasts [40], are all consistent with the hypothesis that ELFs in the meninges of MS patients have a major role in the disease.

These findings have been controversial. On the one hand, the presence of ELFs in the meninges was confirmed by a Howell et al. in London using postmortem brains from 123 SPMS patients from the UK Multiple Sclerosis Tissue Bank [41]. Pikor et al. [20] also found substantial meningeal aggregates in MS brains, labelled them tertiary lymphoid tissue (TLT), and concluded that the meningeal compartment is important in neuroinflammatory processes and that non-immune cells such as stromal cells are also contributory as cellular scaffolds. In contrast, Kooi et al. [42] in a study of brains from the Netherlands Brain Bank could not confirm the presence of ELFs or that meningeal inflammation was associated with cortical demyelination. Peferoen et al. [43] found prominent B-cell infiltrates but did not identify B-cell follicles as defined by the presence of CXCL13 or podoplanin. Torkildsen et al. [44] found B cells in the meninges but no follicle-like structures. In a study of cortical demyelination in early MS, Lucchinetti et al. called the meningeal inflammation “aggregates” of lymphoid cells but did not comment on organization consistent with ELFs [45]. The difference between ELFs or TLTs vs. aggregates may be more important than just nomenclature, since ELFs may be harder to disperse in MS treatment than aggregates. ELFs, meningeal aggregates, and their role in cortical demyelination continue to be an area of active neuroimmunological research.

### 1.6. CXCL13 and the Blood-CSF Barrier

The concentration of CXCL13 in the CSF depends on two processes: diffusion from the serum and intrathecal production. Diffusion of proteins from blood into CSF adheres to the laws of diffusion as a function of molecular size; thus, the concentration quotients, i.e., Q_analyte_ as defined by (analyte concentration_CSF_/analyte concentration_serum_) is higher for small molecules like chemokines and lower for larger molecules like albumin and immunoglobulin G [46]. Albumin is not produced in the CNS, and thus the Q_albumin_ is a measure of blood-CSF barrier integrity. Thus, the mean and median Q_albumin_ in our experience for CSFs and sera from patients with non-inflammatory neurological patients was 0.005, and this figure in MS was similar, indicating that MS patients generally do not have global blood-CSF barrier dysfunction. In contrast, the Q_CXCL13_ in patients with non-inflammatory neurological patients was 0.03, a number that would be predicted from the Q_albumin_ based on the molecular weight of CXCL13 being about 1/7 that of albumin. In clinical situations such as LNB in which intrathecal production of CXCL13 is exuberant (see section of LNB below), with the intrathecal production dwarfing the concentration added by diffusion from the serum, production of CXCL13 can be considered “compartmentalized” [47]. However, the levels of CSF CXCL13 in MS are less than 1/10 those found in LNB, and the contribution to the CSF levels from passive diffusion need to be taken into account when determining whether there is a significant contribution from intrathecal production [48]. Thus, intrathecal production of CXCL13 (IP_CXCL13_) can be estimated by the use of the CXCL13 index (Q_CXCL13_/Q_albumin_) in a similar manner to how the IgG index (Q_IgG_/Q_albumin_), a commonly used measure in clinical assessment of MS patients [49], estimates intrathecal production of IgG. Thus, just like CSF IgG levels on their own cannot be used to estimate IP_IgG_, and the IgG index is required, similarly CSF CXCL13 levels on their own cannot be used to estimate IP_CXCL13_, and the CXCL13 index is required.

### 1.7. CXCL13 as a Clinically Useful Biomarker

#### 1.7.1. Lyme Neuroborreliosis and Other Neurological Infections

CXCL13 has been known as a strongly upregulated biomarker since it was originally noted to be expressed at high levels in the muscles of non-human primates infected with Borrelia burgdorferi [50]. Subsequent work by Narayan et al. [5] revealed that human peripheral blood mononuclear cells (PBMCs), stimulated by Borrelia burgdorferi sonicate, produced CXCL13 and IgG; magnetic separation of PBMC populations and flow cytometry showed that CXCL13 in vitro is produced by dendritic cells. Because LNB in Europe can be difficult to diagnose, the strong upregulation of CXCL13 in the CNS, as detected by CSF CXCL13 levels, has been increasingly used in the diagnosis of LNB in Europe [51,52,53]. Because these levels are so high, relatively insensitive but inexpensive techniques such as ELISA can be used for quantitation (see the Technologies section below). The Euroimmun ELISA (Lubeck, Germany) is approved by the European Medicines Agency (EMA) to test for CXCL13 in the CSF of patients with suspected LNB. However, very high levels of CSF CXCL13 are not specific for LNB and can be found in other infections such as cryptococcal meningitis, neurosyphilis, and AIDS.

#### 1.7.2. Multiple Sclerosis (MS)

In contrast to LNB where the CSF CXCL13 concentrations are very high, i.e., considered positive for the diagnosis if over 250 pg/mL, the concentrations in the CSF of MS patients are usually between 5 and 100 pg/mL. Multiple groups have identified elevated CSF CXCL13 levels in MS patients as a useful biomarker for predicting the disease course [54,55,56,57,58].

The first indication that we could confirm the above work in our laboratory came when we utilized a 40-analyte Luminex panel to determine cytokine/chemokines in the CSF and sera of MS patients and controls [59]. We found that, indeed, CXCL13 was one of most consistently upregulated proteins in MS and was also not elevated in non-inflammatory neurological controls.

In the next series of experiments [60], we focused on CXCL13, using a single molecule Luminex approach, and researched the clinical utility of measuring IP_CXCL13_ in MS, utilizing the CXCL13 index as a measure of IP_CXCL13_. We obtained matched serum and CSF samples from 67 non-inflammatory controls and 67 MS patients and analyzed electronic medical records to determine the level of MS clinical and imaging activity after the LP. As a single predictor of disease activity, the CXCL13 index outperformed CSF CXCL13 levels, the ratio of CSF/serum CXCL13 concentration (i.e., Q_CXCL13_), CSF neurofilament light levels, and oligoclonal bands with respect to sensitivity, specificity, and positive and negative predictive value. Interestingly, combining CXCL13 index and CSF NfL status improved sensitivity and predictive values.

In the next series of experiments [61], we focused on patients with clinically isolated syndrome (CIS). CIS can either be monophasic, with no future attacks or new lesions, or patients with CIS can go on to develop MS that can be debilitating. This is an especially crucial time for making treatment decisions, since early treatment can lower the risks of future episodes in CIS patients and improve long term outcomes. Conversely, there appears to be no benefit to treating CIS patients who will remain monophasic. CIS thus represents a “window of opportunity” for optimal treatment with DMTs, but the optimal strategy is controversial, and what is urgently needed is a biomarker with a high negative and positive predictive value. The challenge is to identify those CIS patients at higher risk of future disease activity, and the evidence from the earlier studies indicated that the CXCL13 index might be an excellent biomarker for this. In patients with clinically isolated syndrome (CIS) or radiologically isolated syndrome (RIS) that were followed for at least 5 years after the LP, patients with high CXCL13 indices were more likely to convert to clinically definite MS (82.4%) compared to those with low CXCL13 indices (10.0%).

In our most recent work [62], we looked at the potential use of the CXCL13 index for guiding treatment of CIS. Patients with clinically isolated syndrome (CIS) were divided into those with low or high CXCL13 indices. A low CXCL13 index predicted that a ME-DMT or no DMT would be successful in achieving NEDA (no evidence of disease activity), while a high index predicted failure of ME-DMTs to achieve NEDA. The CXCL13 index outperformed other potential predictive biomarkers in CIS. The data from this study indicate that patients with CIS do well with ME-DMTs or no DMTs if the CXCL13 index is low, but they fail treatment with ME-DMTs if the I_CXCL13_ is high.

The CXCL13 index has also been used by Alvarez et al. [63] to identify optimal responders to the B-cell depleter rituximab. This group also found that CXCL13 levels in the CSF dropped dramatically with rituximab therapy. In addition, Fissolo et al. [64] have demonstrated that serum CXCL13 levels significantly decreased after treatment with teriflunomide in NEDA compared with EDA patients after 1 year of treatment.

Regarding lymphoma, CXCL13 levels can be very high in B-cell lymphomas, possibly related to production of CXCL13 by malignant B cells [65]. Because of this, CSF CXCL13 levels in primary CNS lymphoma, a diagnosis that can be difficult to make, can be very high and decrease with therapy; thus, CSF CXCL13 can potentially be utilized to assist in diagnosis and prognosis [66,67].

Considering inflammatory/autoimmune diseases, serum CXCL13 concentrations have been found to be elevated in Sjogren’s syndrome [68,69] and in systemic lupus erythematosus with disease activity [70,71], and CSF CXCL13 levels have been proposed as potentially useful biomarkers of treatment response and outcome in autoimmune encephalitis [72,73].

## 2. Technical Considerations for the Measurement of CXCL13

### 2.1. Technologies Available for the Measurement of CXCL13 Protein: Well- and Bead-Based Assays, Point-of-Care Assays, and Oligonucleotide/Proximity Linking Assays

Considering technologies available for the measurement of CXCL13 protein, there are many advantages in using CXCL13 as a biomarker in human diseases, one of them being the number of methods are available for its measurement. A variety of methods have been used by investigators over time ever since the first description of this chemokine 26 years ago, each with its advantages and disadvantages as outlined below and in the Table 1. Another advantage of CXCL13 as a potential biomarker is the fact that it is extremely stable over time [74,75]. For some clinical conditions in which levels are very high, such as LNB or lymphoma, fast and inexpensive but relatively insensitive methods such as ELISA or lateral flow immunoassays (LFA) can be used. The methods reviewed below apply mostly to CXCL13 measurement in CSF, but these technologies have also been used for serum or plasma. Most of the methods below cannot be used for immediate diagnosis on patient presentation (point-of-care) since they require analysis in a specialized laboratory, with the exception being LFA assay technology, item (2) below.

#### 2.1.1. Well-Based Immunosorbent Assays

The most commonly used assay system has been the enzyme-linked immunosorbent assay (ELISA), a relatively old, tried-and-true methodology, first described in 1971 [76], and the most commonly used kits have been those available from Euroimmun (Lubeck, Germany) and Quantikine (R&D Systems, Minneapolis, MN, USA). The laboratory hardware required for the assay is only an ELISA reader which is found in almost all clinical and research laboratories. Many laboratories also use automated plate washers, but the washing can be done by hand.

#### 2.1.2. Point-of-Care Testing

In parts of the world where LNB is not rare, and the diagnosis can be difficult, the ability to diagnose the infection immediately on presentation is helpful. Thus, Finnish and German investigators have published studies of point-of-care testing for LNB using the lateral flow immunoassay [77,78]. LFA assays are relatively insensitive, but given the very high levels of CXCL13 in the CSF in LNB, i.e., sometimes over a thousand pg/mL, sensitivity for the LFA assays is not necessary; however, false negatives can be a problem. This assay is similar to LFA assays for COVID [79] which have become widespread, and it allows easy and fast diagnosis of this infection.

#### 2.1.3. Bead-Based Immunosorbent Assays

In contrast to the relatively simple and inexpensive technologies described above, bead-based assays, such as Luminex and single molecule array (SIMOA), are more complex, are more expensive when assaying single analytes, and require highly specialized laboratory hardware. However, they are much more sensitive and can be used to measure multiple analytes if required. Our group has used both ELISAs and Luminex to measure CSF and serum human CXCL13 levels, and our experience will be summarized below. Other groups have used SIMOA [15,64,80], which is even more sensitive than Luminex, but it requires even more specialized hardware. Neither Luminex nor SIMOA capability is widespread in clinical and research laboratories at this point in time, but their use is increasing as the need for more and better biomarkers expands.

To demonstrate the increasing sensitivity with increasing complexity of these methodologies, a comparison of their relative lower limit of quantitation (LLOQ) is instructive. A common definition of this measure of sensitivity is the lowest concentration at which the coefficient of variation is <20%. For the three assays (ELISA, Luminex, and SIMOA) the LLOQs are, respectively, 10, 0.7, and 0.07 pg/mL. For CSF CXCL13 quantitation in MS, many investigators have found the poor sensitivity of the ELISAs problematic. Hytonen et al. [81] found, in a group of 34 patients with MS or other neuroinflammatory disease, all of whom had positive CSF oligoclonal bands, that 24 (71%) had undetectable CSF CXCL13 levels by using the Quantikine ELISA. In contrast, Olesen et al. [80] were able to quantitate CSF CXCL13 in all 63 optic neuritis patients using SIMOA, compared to only 10/37 (27%) when the same group used the Euroimmun ELISA [82].

#### 2.1.4. Oligonucleotide/Proximity Linking Assays

This methodology is relatively new. It utilizes a combination of antibody specificity and PCR amplification to enhance sensitivity. Akesson et al. [83] utilized Olink to develop a combination of 11 proteins in the CSF to predict the severity of disability worsening; one of those proteins was CXCL13. Unfortunately, this manuscript used a protein measure called “normalized protein expression (NPX)” determined by the Olink data analysis software, and there were no CXCL13 concentration levels in pg/mL available. Thus, at this point, this methodology has unknown utility in clinical MS, but further research may prove it is an exciting option, not only for CXCL13, but also for possible lower concentration biomarkers that require the ultimate in sensitivity.

### 2.2. Methodological Issues in the ELISA and Luminex Assays

Our group has used ELISA and bead-based assays for the measurement of CXCL13 in CSF and sera. The assays of serum CXCL13 were straightforward and yielded reproducible results, partly because levels in the sera in MS patients and controls were moderately high, usually between 20 and 400 pg/mL with a median of 79 pg/mL. These levels were generally below those found in the CSF of LNB patients but well within the linear portion of the standard curves of both ELISA and bead-based assays.

The story for CSF CXCL13 levels in MS patients was different. Early in our work on measuring we had the advantage of having available results from an excellent comparative study of ELISAs and bead-based assays for CSF CXCL13 in LNB by Henningsson et al. [84]. They warned that cut-off levels were difficult to standardize and that different laboratories may get different results. In a similar study of 100 patients with definite or possible LNB or controls, Markowicz et al. [85] found different cutoffs than those published in the Henningsson study, with a cutoff for diagnosis of LNB of 131 pg/mL for the Mikrogen bead-based assay and 259 pg/mL for the Euroimmun ELISA.

Given the well-known problem with sensitivity of ELISAs for CXCL13 and the low levels of CXCL13 in the CSF in MS patients relative to LNB, we proceeded to attempt to develop a Luminex assay, the SinglePlex assay for CXCL13 available from Bio-Rad (Hercules, CA, USA) in the hope that we would achieve improved sensitivity. In fact, the quality control for this assay was performed by the company in a serum-based matrix, which the specification sheet advertised as a lower limit of quantitation (LLOQ) of 0.7 pg/mL. The samples for the construction of the standard curve for this assay were recombinant CXCL13 proteins. Since the levels of CXCL13 are usually lower than serum, which is added to the wells at a 1:4 dilution, the CSF is added neat, which raises the possibility that matrix effects in appropriate diluent selection may be a problem. Optimal diluent selection for standards is a common challenge in bead-based assays [86].

We found that we could not reliably use this assay off-the-shelf, with the provided standards and standard diluent, for CSF CXCL13, and determined that the most likely problem was indeed matrix effects. Thus, the readout of the assay for each well is a “mean fluorescent index” (MFI), which in repeated measures of the CSFs was consistent, with a low coefficient of variation (CV). The problem was that the recombinant standard curve was highly dependent on matrix: very high MFIs for standards in 0.05% Tween, moderately high MFIs for standards in the “standard diluent” provided in the kit, and low MFIs for standards in fluids resembling human CSF (aCSF, PBS-BSA, pooled normal CSF). We were able to solve this problem when we found that matrix effects were minimal when standards containing human, rather than recombinant, CXCL13 were used. When human standards were used, either PBS or aCSF, which mirror human CSF, could be used as a diluent. Since the Mikrogen assay is a bead-based assay for CSF CXCL13 used to diagnose LNB in Europe, we tried to identify the composition of the standard, but we were unable to obtain that information; we feel it is likely to be a human CXCL13.

## 3. Conclusions

CXCL13 is a key chemokine in the immune system and is important for the trafficking of B cells and follicular helper T cells. Increasing evidence points to its utility as a biomarker, especially early in the management of MS patients when treatment decisions are highly complex, and measuring intrathecal production of CXCL13 can potentially guide optimal therapy.

## Figures and Tables

**Table 1 biomolecules-14-01541-t001:** Comparison of methods available for CXCL13 measurement.

Methodology	Sensitivity	Availability	Cost
Well-based (ELISA)	+	+++	+++
Bead-based (Luminex or SIMOA)	+++	++	++
Point-of-care	+	+	++
OLINK	unknown	+	+

Legend: Assay technologies ranked with +++ = advantage, ++ = neither advantage nor disadvantage, + = disadvantage.

## Data Availability

No new data were created or analyzed in this study. Data sharing is not applicable to this article.

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
