# Peer review of "CXCL13 as a Biomarker: Background and Utility in Multiple Sclerosis"

_biomolecules, 2024, doi:10.3390/biom14121541_

Round 1

Reviewer 1 Report

Comments and Suggestions for Authors

The authors carry out an extensive review on CXCL13 as a potential biomarker of inflammation in different disease, mainly LNB (where the level are the highest) and MS. There is also an extensive paragraph on more technical issues on CXCL13 measurements.

I have a comment: there are studies which investigate the possible role of CXCL13 as a biomarker of response to therapy? Moreover, I would integrate the paragraph of MS with more studies about the role in predicting the disease other than the studies already conducted by the authors.

A correction :

Page 5: line 242, there is a paragraph starting with "3." but "2" and "1" is lacking, this is misleading for the reader. I suppose you want to list the diseases in which CXCL13 is involved. Please correct, reformulate the paragraph if needed.

Author Response

  1. “possible role of CXCL13 as a biomarker of response to therapy”-

            i.The review already has the following: “The CXCL13 index has also been used by Alvarez et.al. 63 to identify optimal responders to the B-cell depleter rituximab. This group also found that CXCL13 levels in the CSF dropped dramatically with rituximab therapy.”

            ii.To amplify this role of CXCL13, I have included the following in the revised manuscript: “In addition, Fissolo et.al. have demonstrated that serum CXCL13 levels were significantly decreased after treatment with teriflunomide in NEDA compared with EDA patients after 1 year of treatment.”

2.“ I would integrate the paragraph of MS with more studies about the role in predicting the disease other than the studies already conducted by the authors.”

            The following sentence is already contained in the MS section:” Multiple groups have identified elevated CSF CXCL13 levels in MS patients, 54 55 56 57 58 raising hopes that CXCL13 can be used as a biomarker in this unpredictable disease.” In the revised manuscript, I have revised this statement to the following:” Multiple groups have identified elevated CSF CXCL13 levels in MS patients as a useful biomarker for predicting the disease course 54 55 56 57 58 . “

3.“Page 5: line 242, there is a paragraph starting with "3." but "2" and "1" is lacking, this is misleading for the reader.”

I have revised the headings and included an outline at the beginning for clarification.

Reviewer 2 Report

Comments and Suggestions for Authors

The manuscript addresses an interesting and scientifically important topic of CXCL13 as a biomarker in neurological diseases, especially in multiple sclerosis (MS). The choice of presented issues (encompassing practical aspects) adds significantly to the potential informative value and novelty of this review. However there are some problems which have to be resolved.

In certain parts of the manuscript the authors quite frequently repeat the exact words and even extensive phrases of other authors. Although put in quotation marks and properly cited, it would be more suitable to formulate an own description and/or opinion based on the reviewed topic.

In the section entitled: “CXCL13 as a clinically useful biomarker”, subsection “Multiples sclerosis” the authors described almost exclusively their own results with very little information regarding the findings of other groups which is not satisfactory in the review article focused on MS.

The methodological part of the manuscript represents definitively a potentially great value for the readers but it should be reorganized, e.g. by moving a part of the information into a table summarizing advantages/disadvantages and practical considerations of particular methods

There are also some language mistakes such as: “Another viral infection …. is  mouse hepatitis virus(MHV)”,  or “the main production of CXCL10 … was astrocytes” or superfluous sentences like in the section “Point-of-care testing”:  “LFA assays are relatively insensitive but given the very high levels of CXCL13 in the CSF in LNB, i.e. sometimes over a thousand pg/ml, sensitivity for the LFA assays is not necessary. …. A disadvantage of LFA assays is their insensitivity, which results in a fairly high level of false negatives, unless the analyte concentration is very high.”

The manuscript has to be also carefully edited e.g. due to the inconsistent numbering/marking of sections and sub-sections.

Author Response

1.“In certain parts of the manuscript the authors quite frequently repeat the exact words and even extensive phrases of other authors. Although put in quotation marks and properly cited, it would be more suitable to formulate an own description and/or opinion based on the reviewed topic.”

            I have followed Reviewer 2’s recommendation for all phrases, with only one exception, the definition of Aloisi and Zhan for ELFs because I felt was important that the manuscript uses the precise definition as it was worded in the manuscript.

2.” In the section entitled: “CXCL13 as a clinically useful biomarker”, subsection “Multiples sclerosis” the authors described almost exclusively their own results with very little information regarding the findings of other groups which is not satisfactory in the review article focused on MS.”

            This is identical to criticism #2 from Reviewer 1- see my response(above)

3.”The methodological part of the manuscript represents definitively a potentially great value for the readers but it should be reorganized, e.g. by moving a part of the information into a table summarizing advantages/disadvantages and practical considerations of particular methods.”

            I have followed the reviewer’s recommendation and created a table summarizing advantages and disadvantages of the methods.

4.”There are also some language mistakes such as: “Another viral infection …. is  mouse hepatitis virus(MHV)”,  or “the main production of CXCL10 … was astrocytes” or superfluous sentences like in the section “Point-of-care testing”:  “LFA assays are relatively insensitive but given the very high levels of CXCL13 in the CSF in LNB, i.e. sometimes over a thousand pg/ml, sensitivity for the LFA assays is not necessary. …. A disadvantage of LFA assays is their insensitivity, which results in a fairly high level of false negatives, unless the analyte concentration is very high.””

            The manuscript has been reviewed and language mistakes corrected, including those listed above.

5.”The manuscript has to be also carefully edited e.g. due to the inconsistent numbering/marking of sections and sub-sections.”

            This criticism is similar to criticism #3 from Reviewer 1. I have edited the manuscript, and revised the numbering of the sections and sub-sections, and included an outline to make the organization clearer.